# Trade-Off between Task Accuracy, Task Completion Time and Naturalness for Direct Object Manipulation in Virtual Reality

Jari Kangas [1,*,†]  , Sriram Kishore Kumar [1,†], Helena Mehtonen [2], Jorma Järnstedt [2] and Roope Raisamo [1]

1   Faculty of Information Technology and Communication Sciences, Tampere University,
    33014 Tampere, Finland; sriramkumar.kishorekumar@tuni.fi (S.K.K.); roope.raisamo@tuni.fi (R.R.)
2   Medical Imaging Centre, Department of Radiology, Tampere University Hospital, 33520 Tampere, Finland;
    helena.mehtonen@pshp.fi (H.M.); jorma.jarnstedt@pshp.fi (J.J.)
*   Correspondence: jari.a.kangas@tuni.fi
†   These authors contributed equally to this work.

**Abstract:** Virtual reality devices are used for several application domains, such as medicine, entertainment, marketing and training. A handheld controller is the common interaction method for direct object manipulation in virtual reality environments. Using hands would be a straightforward way to directly manipulate objects in the virtual environment if hand-tracking technology were reliable enough. In recent comparison studies, hand-based systems compared unfavorably against the handheld controllers in task completion times and accuracy. In our controlled study, we compare these two interaction techniques with a new hybrid interaction technique which combines the controller tracking with hand gestures for a rigid object manipulation task. The results demonstrate that the hybrid interaction technique is the most preferred because it is intuitive, easy to use, fast, reliable and it provides haptic feedback resembling the real-world object grab. This suggests that there is a trade-off between naturalness, task accuracy and task completion time when using these direct manipulation interaction techniques, and participants prefer to use interaction techniques that provide a balance between these three factors.

**Keywords:** 3D visualization; virtual reality; object manipulation; hand tracking; controller interaction

## 1. Introduction

Virtual reality (VR) devices, e.g., head-mounted displays (HMDs), are becoming mainstream tools in a wide range of different domains, for example, in entertainment, marketing, education, and training. They make it easy to cater immersive experiences with the user situated in a realistic three-dimensional (3D) environment where he/she can, for example, observe, learn and act on virtual, dynamic constructions that could not be arranged in reality.

VR devices have been introduced also into the medical field, especially in visualization and training applications [1]. For example, Gunn et al. [2] studied the impact of a VR learning environment on the development of technical proficiency for medical imaging (diagnostic radiography). In addition, Reymus et al. [3] found that students appreciated the VR simulation over two-dimensional radiographs in learning root canal anatomy, and Ryan et al. [4] reported that virtual learning significantly improved students' satisfaction/engagement and recall. Almiyad et al. [5] presented an augmented reality (AR) tool to help doctoral students learning percutaneous radiology procedures.

Currently, in medical diagnosis and operation planning, clinicians perform their task of observing the 3D models on two-dimensional (2D) screens using 2D software. The clinicians use 2D interfaces, such as a mouse and keyboard, to translate and rotate the 3D model to observe it from different angles. However, using 3D environments to perform the same task will provide better understanding [6,7]. Due to medical risks involved, the

clinicians require a reliable interaction technique and interface to precisely manipulate 3D objects in VR.

Rigid object manipulation (such as the model observations, above) is one of the fundamental tasks in virtual reality besides navigation and way finding, according to Bowman and Hodges [8]. Previous research results indicate (e.g., [9–11]) that hand-tracking-based systems are not necessarily as accurate or efficient as controller-based systems for direct object manipulation in VR. However, if hand tracking is a viable interaction method, the system user would be able to avoid learning new control methods. Two direct manipulation interaction techniques, *HandOnly* (pinch gesture) and *Controller+Trigger*, differ in terms of two design parameters: the tracking method and action for object pickup. To evaluate the designs, we included a third interaction technique, which combines the controller tracking with hand gestures. This hybrid interaction technique called *Controller+Grab* is based on the functionality of the Valve Index controller [12] with finger tracking and makes use of the grab gesture.

A controlled experiment was carried out using commercially available interaction technologies with 12 participants, using within-subject design, and the conditions were presented in balanced order. We measured objective measures, such as the task completion time, position and orientation error and subjective measures of confidence, ease of use, hand tiredness, naturalness (defined by Navarro and Sundstedt [13] as an interaction concept of "using your own body"), and using daily. The results demonstrate that the hybrid interaction technique was found to be the most liked, as it was intuitive, easy to use, fast, reliable and it provided haptic feedback resembling the real-world object grab.

Wagner et al. [14] compared hand interaction, controller interaction and their combination for data manipulation task in a virtual reality environment. Similar to our experiment, the results indicate that while there were no significant differences between methods in task performance or workload, the participants preferred the mixed mode.

Earlier studies (e.g., [9–11,13,15,16]) compared hand-tracking- and controller-based interaction techniques in terms of task accuracy and task completion time and used these two factors to choose the optimal interaction technique for object manipulation. The present experiment demonstrates that there is a trade-off between naturalness, task accuracy and task completion time when using these direct manipulation interaction techniques, and we should also consider naturalness while making the decision of choosing the interaction technique for direct object manipulation. This implies that future direct manipulation interaction technique designs should consider these three factors: task accuracy, task completion time and naturalness.

In summary, the contributions of this paper are (1) an evaluation of three interaction techniques for direct manipulation of a rigid object, (2) finding of the trade-off between different factors for direct manipulation techniques.

## 2. Background

### 2.1. Input Methods for Precise Object Manipulation in VR

Previous studies have explored input devices such as hands, controllers, data gloves, mouse and pen for object manipulation in VR.

Hand tracking [17–22] is a prime example of direct manipulation methods (Shneiderman [23]) in which objects are interacted physically and incrementally with immediate feedback. While hand tracking is usually considered a preferred and natural interaction method, a well working and robust hand pose recognition technology for VR applications has been difficult to develop. The introduction of Leap Motion [24] enabled convenient and relatively accurate hand tracking for VR applications. One possible technical solution to improve hand tracking accuracy would be to use data gloves (e.g., [17,25–28]), which usually have dedicated sensors to obtain more accurate position and pose data. Wearing the data gloves, on the other hand, may feel unnatural to some users because of the required constructions. Force feedback devices are used for medical

training and operations (e.g., [29–31]) and would provide very accurate position and pose data, but holding the linked device would restrict the space of possible moves.

Several indirect manipulation techniques, such as handle bar [32], smart pin [33], crank handle [34], mid-air objects on rails (MAiOR) [35], degrees-of-freedom (DoF) separation widget [36], decomposition and integration of degrees of freedom (DIOD) [37] have been designed for indirect manipulation. These indirect manipulation techniques are either more precise [36,37] or fast to use [32,37]. These interaction techniques provide precision to the user by using strategies, such as DoF separation [33–37], two handed control [32,34], Control Display [35], rotation-translation separation [33–37]. However, these techniques suffer from either hand tiredness [32] or increased learning curve [35,36] which are important for medical context. Thus, we focused on direct manipulation techniques, as these are straightforward for users to learn and use. The focus of our research was to identify an inexpensive input device appropriate for direct manipulation, which is both precise and fast, while being easy to learn and natural.

There are several recent studies where hand tracking methods (in many cases based on Leap Motion, e.g., [9–11,13,15,16]) were compared against controller-based interaction methods. In most cases, hand tracking compared unfavorably against the regular controllers for object manipulation tasks [9–11,13,15,38] which require precision. Masurovsky et al. [15] compared the standard and modified Leap Motion-based hand tracking against an Oculus controller and found the controller solution to outperform both hand tracking alternatives. Caggianese et al. [9] studied a VR system where a person was asked to manipulate a set of boxes in various tasks (for example, to build a tower of boxes) using either an HTC Vive controller [39] or bare hands (recognized by a Leap Motion sensor). The completion times were significantly faster with the controller than with the hand interface. Gusai et al. [10] compared the controller and hand tracking in a collaborative setting for the object placement task and concluded that controllers give better performance in accuracy and in usability. Galais et al. [11] compared the controller and hand tracking in a cube manipulation task (cubes arranged in a given order) and observed a significantly higher cognitive load and lower performance for hand tracking than for the controller use. Navarro and Sundstedt [13] compared the Vive controller and hand tracking in the manipulation of objects in two VR games, a Pentomino puzzle game and ball throwing game. They reported a significant decrease in player performance when using hand tracking, and that hand gestures were not as reliable as the Vive controller.

In some studies [40,41], data gloves and pen devices were preferred over the controller. Lee et al. [40] compared controller- and data glove-based hand gesture interfaces for VR games, expecting the users to prefer the data glove interface. Pham and Stuerzlinger [41] compared a mouse, a pen device and a controller in VR and AR selection tasks. They demonstrated that the pen device outperformed the controller in speed and was more liked.

## 2.2. Input Methods for Imprecise Object Manipulation

The above input methods were also compared for manipulation tasks in virtual reality which do not focus on precision. Tasks used in these studies include data exploration, opening objects and grasping. Similar to precise object manipulation, controllers performed better than bare hands for imprecise object manipulation tasks. Huang et al. [42] evaluated two-handed hybrid interaction methods, where all four combinations of using either hand tracking or a controller in each hand were tried in four different tasks. The results showed that the interaction system with controllers in both hands was the most efficient. In tasks that we studied, the use of two hands is unnecessary when the task can easily be executed by one hand. Liu et al. [43] demonstrated their data glove design and showed that it compared positively for the grasp capability against the Leap Motion-based hand-tracking system.

However, there are some studies that showed that participants prefer hands or that there is no difference between hands and controllers. Figueiredo et al. [44] compared hand tracking and controller interaction on a set of scenarios and found them equally efficient.

Their results indicate that while in some tasks (e.g., selection), the controller was faster and more accurate, hand tracking was subjectively liked by the participants.

Reski and Alissandrakis [38] compared gamepad, VR controller and hand gestures in an open data exploration task in a virtual reality environment. Their results did not show any significant differences between the interaction methods on the user experience.

### 2.3. Object Manipulation Techniques for Medical Domain

In the medical field, a 2D monitor with a keyboard and a mouse interface is traditionally used for operation planning, which requires one to make a mental 3D model of the anatomical structures. Due to the lack of 3D perception, the task is complicated, error prone, and requires training. The planning process is often time consuming and requires a high cognitive load (Mendes et al. [45]). Viewing 3D data in a 3D environment (in VR) would provide an advantage over 2D displays for perception and understanding (Sutherland [6]).

Various input methods have been used for object manipulation for operation planning. Xia et al. [46] used a 3D mouse, Hsieh et al. [47] created a special 3D tracker attached to medical tools, and Olsson et al. [48] used a force feedback haptic device. Li et al. [49] compared two haptic VR interfaces (a force feedback device and a handheld controller with vibrotactile feedback) and a 2D interface (a mouse) for the medical marking of 3D models of human anatomic structures and demonstrated their practical usability. The regular 2D mouse does not fit well for VR, as it has only 2 degrees-of-freedom (DoF) which is incompatible with the VR interaction requirements [50–52]. Thompson [52] compared a hand gesture interface and 2D mouse and found that the gestures had an advantage for object rotation, translation, and scaling.

Input methods have been used also for imprecise object manipulation in the medical domain. Khundam et al. [16] compared hand-tracking-based and controller-based interaction methods for medical training in VR. The results indicate that there are no differences in completion times, SUS evaluations or usability, but the authors consider hand tracking to be more natural (more similar to real operations) and, therefore, preferable. However, the medical training in this study did not require very accurate manipulation of the objects. Fahmi et al. [53] compared a controller, a data glove and hand tracking for anatomy learning and demonstrated how the controller was superior for usability and ease of learning against the two other interaction methods.

## 3. Method

In the experiment, we used a within-participant design and we balanced the order of conditions for different participants using Latin squares to mitigate the learning effect.

### 3.1. Experiment Design

The objective of this experiment was to get a better understanding of how well the selected interaction techniques work for the basic manipulation operations in virtual reality environments. Several studies compared hand gestures with the controller trigger technique. These two interaction techniques differ in terms of tracking and action for object selection. We introduced a hybrid condition for our study, which uses the controller tracking and uses hand gestures for object selection. Thus, the designs of our selected interaction techniques to be compared in our study are described in Table 1. These interaction techniques are visually explained in Figure 1.

Pinch gesture was selected as the hand gesture for *HandOnly* condition, as it was used in previous comparison studies ([9–11,13]) and it is more robust to noisy hand tracking in comparison to power grasp ([13]).

**Table 1.** The three interaction techniques (*HandOnly*, *Controller+Trigger* and *Controller+Grab*) based on two position tracking methods (hand and controller tracking), and two object pick up action types (trigger press and hand gesture).

| | | Tracking Method | |
|---|---|---|---|
| | | **Hand Tracking** | **Controller Tracking** |
| Action for pick up | Trigger Button | -Not possible- | *Controller+Trigger*: Trigger press in controller |
| | Hand Gesture | *HandOnly*: Pinch | *Controller+Grab*: Grab using Valve Index controller |

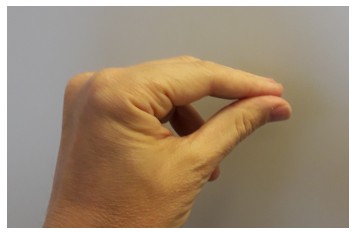 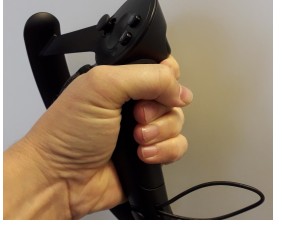 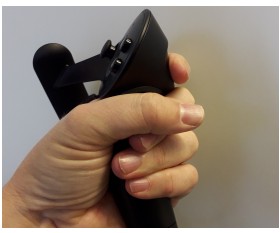

(**a**) *HandOnly*        (**b**) *Controller+Grab*        (**c**) *Controller+Trigger*

**Figure 1.** The interaction mechanisms to pick up the object that were used for different conditions: (**a**) in *HandOnly* condition, the participant uses a pinch gesture, (**b**) in *Controller+Grab*, the participant uses the grab gesture on the Valve Index controller, (**c**) in *Controller+Trigger*, the participant presses the trigger button on the controller.

Based on object selection and manipulation taxonomy by Bowman and Hodges [8], we created a simple task of picking an object from one location, then translating and rotating to another location to match the position and orientation of the target in the VR environment (we defined a threshold for how close to the target location and orientation the cube must be for the action to be acceptable). We made sure that the object was comfortably in reach both in the picking up and in the putting down locations. The experimental system was created such that both hands could be used equally, the cubes were moved either from left to right or from right to left, and either hand could be used to hold the cubes.

We formulated the following research questions to guide our work:

RQ1    Are there any statistically significant differences between the selected interaction techniques in task completion times and placement accuracy?

RQ2    Are there any statistically significant differences between the selected interaction techniques in participant preferences?

For testing purposes, we also had the following hypotheses that we could test with the final data concerning our use case:

**Hypothesis 1 (H1).** *The participants would prefer HandOnly use over the Controller+Trigger use.*

**Hypothesis 2 (H2).** *The task completion times would be shorter for Controller+Trigger use than for HandOnly use.*

*3.2. Measurements*

3.2.1. Objective Measures

The main measure was the task completion time, how long the participant spent in each single task. The time was measured from the start until the participant confirmed the task was done after putting the cube down in an acceptable position and orientation. Secondary measures were the errors in the cube placement. We measured separately the distance error and the orientation error.

The distance error $E_d$ (see Equation (1)) was defined as the Euclidean distance between the target position of the cube and the actual final position.

$$E_d = \frac{|P_t - P_f|}{0.1} \cdot 100,$$  (1)

where $P_t$ is the target position (measured in meters) and $P_f$ is the final position of the cube. The distance error had a value 100 when the object was 10 cm away from the target.

The orientation error $E_o$ (see Equation (2)) was defined as a scaled angle difference between the target orientation vector and the final cube orientation vector when the participant was ready.

$$E_o = \frac{AngDiff(O_t, O_f)}{360} \cdot 100,$$  (2)

where $E_o$ is the orientation error, $O_t$ is the target orientation vector, $O_f$ is the final orientation vector of the cube, and $AngDiff()$ computes the angle (in degrees) between two orientation vectors. The maximum orientation error was a value 50 when the orientations were in totally opposite directions. To make sure that there were no accidental completions of the task, we set a threshold value of 10 such that the average of distance error and orientation error $T_f = (E_d + E_o)/2$ must be below 10 before the participant could complete the task. In practice, that meant that the object must be closer than 10 mm away from the target position.

### 3.2.2. Subjective Questions

After each condition, we asked the participants how they felt about several aspects of the experiment as listed in Table 2 on a 7-step Likert scale for the answers, where 1 represents completely disagree and 7 represents completely agree. For questions S4 and S5, the participants were asked to provide reasoning behind their ratings.

**Table 2.** The questions S1 to S5 that were asked to evaluate the subjective impressions of each individual interaction method.

| | |
|----|------------------------------------------------------------|
| S1 | How confident were you in your ability to use the interaction method? |
| S2 | How easy/effortless was the interaction method to use? |
| S3 | How tired are your hands? |
| S4 | How natural was it to use? |
| S5 | Could you imagine using the method daily? |

After completion of the conditions, we evaluated how the interaction methods compared to each other by asking the participants to provide a ranking of the methods (1st, 2nd, and 3rd) in three aspects as listed in Table 3. For all questions, we also asked for a short description as to why the participant made that specific ranking.

**Table 3.** The questions R1 to R3 that were asked to rank the interaction methods.

| | |
|----|------------------------------------------------------------|
| R1 | Which interaction method did you like the most? |
| R2 | Which interaction method did you think would be the easiest to adopt by a novice? |
| R3 | Which interaction method had the most potential if it could be developed further? |

### 3.3. Statistical Significance Measures

We used a Monte Carlo permutation test [54–57] to analyze possible statistically significant differences between different parameters sets. The permutation test is not dependent on as many assumptions on the sample distribution as some other tests such as ANOVA [54]. Especially important is that the test sample need not be normally distributed.

Additionally, we were using median values in most cases as the test statistic, while some other methods can only use the mean. Compared to the mean, the median is more tolerant to outliers in the data.

In all tests, an observed value of a measurement is compared against a distribution of measurements produced by resampling a large number of sample permutations assuming no difference between the sample sets (null hypothesis). The relevant *p*-value is then given by the proportion of the distribution values that is more extreme or equal than the observed value. To get the distribution of measurements assuming no difference between the conditions, we randomly switched the participant samples between the conditions, generating 10,000 permutations to be measured.

## 4. Experiment

### 4.1. Apparatus

#### 4.1.1. Hardware Components

The experimental system was built using the Varjo VR-2 Pro HMD [58] and relevant accessories. The Varjo device has an integrated hand-tracking system. For controllers, we used the Valve Index controllers [12], which track controller position using Vive base stations V2.0 and track individual finger positions, using capacitive and force sensors.

#### 4.1.2. Software Components

The experimental system was built using the Unity 3D software development system [59] and the Varjo SDK [60].

### 4.2. Pilot Study

We arranged two pilot tests, where the parameters for the experimental system were fine tuned. In the pilots, we made sure that the arrangement is functional, and the locations are easily reachable. The main question to clarify was the number of repeats of each task, as we did not want to tire the participant unnecessarily but still obtain enough data. In the end, we had 6 tasks for each interaction method, making a total of 18 tasks. The whole experiment, including a short practice session before each interaction method, was to take around 30 min.

### 4.3. Participants

As for the participants, we sent invitations to local university mailing lists and recruited 12 people, both students and personnel. The ages varied from 22 years to 47 years, the mean age was 33.0 years. There were 7 (out of 12) female participants. Eleven participants had some experience in using VR previously, the most common experience being one year and average being two years.

### 4.4. Procedure

Upon arrival, the participant was introduced to the study and to the equipment. The participant was asked to read and sign a consent form, and fill in a background information form. The participant then put on the head-mounted display, and the facilitator started the first condition session (see Figure 2).

For each condition, the participants were given enough time to practice interaction techniques until they were confident. In the practice session, the cube was reset to the fixed start position when the done button was pressed, and they were able to practice multiple times. The session started by the participant by pressing a "start" button and was run automatically from the beginning to the end. In the beginning of each task, the participant could see two cubes as shown in Figure 3. The green cube with orange 'T' is the cube that the user can manipulate and represents the start location for the task. The transparent grey cube with green 'T' is the target end location where the participant has to move the green cube and align it. The participant could confirm the task completion by pressing the "done" button after which a new cube appeared in the start location. This process was repeated six

times for each condition. Three tasks involved moving the cube from left to right and the other three from right to left. The distances between the start and end locations varied, as well as the orientation of the cube in the target location.

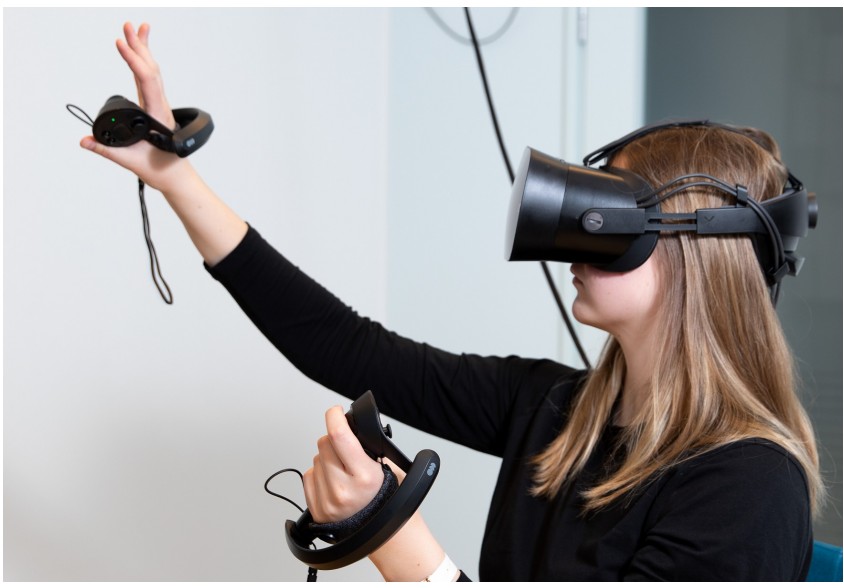

**Figure 2.** The participant sat with the headset on and manipulated the virtual cubes either by holding the controllers or with bare hands. In this example, the participant is about to pick an object with her right hand in the *Controller+Grab* condition.

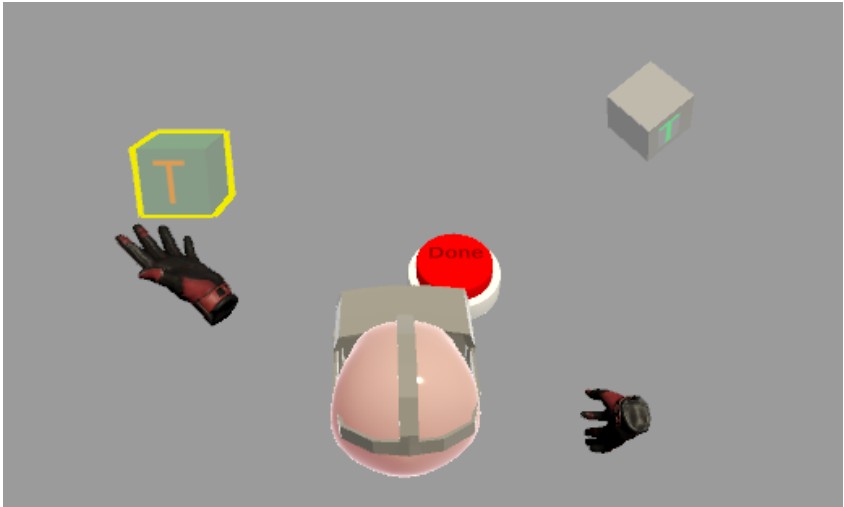

**Figure 3.** The initial setup as seen by a participant inside the VR environment

After each condition the participant removed the HMD and filled in a post-condition evaluation form. After going through all the experiment conditions, the participant was instructed to remove the HMD. He/she was then asked to fill in the post-experiment evaluation form.

## 5. Results

### 5.1. Task Completion Times

The completion times are visualized in Figure 4 (left). We found statistically significant differences between the interaction techniques, where *HandOnly* was significantly slower than both other methods (the $p$-value was in both cases below $0.017 (= 0.05/3)$, using Bonferroni correction of doing three comparisons, $p < 0.005, CI = 5.6 - 12.4, 1 - \beta = 0.92$ for comparison between *HandOnly* and *Controller+Trigger*, $p < 0.005, CI = 4.7 - 10.9, 1 - \beta = 0.86$

for comparison between *HandOnly* and *Controller+Grab*, where $1 - \beta$ is the statistical power of the test). There was no significant difference between the *Controller+Trigger* and *Controller+Grab* methods. The VR experience did not indicate any correlation with the completion times.

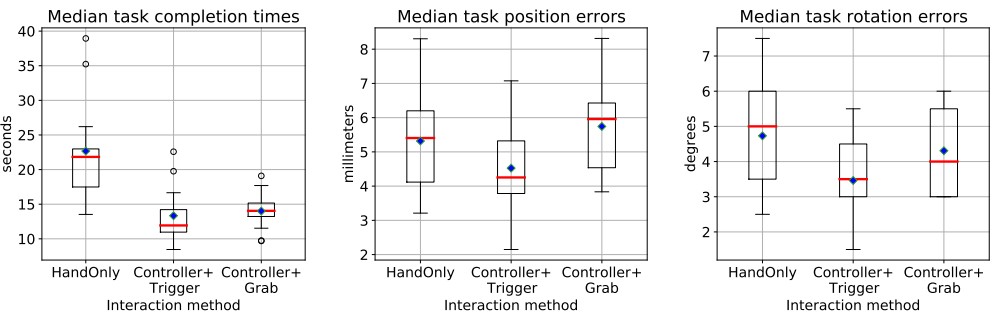

**Figure 4.** The task completion times (**left**), the object position errors between the start and target positions (**middle**) and the object rotation errors between the start and target orientations (**right**). For each participant, we computed the median values over six tasks. The median values over all participants are drawn as red lines and the mean values as blue diamonds.

### 5.2. Position and Orientation Errors

The position and orientation errors are visualized in Figure 4 (middle and right). The only statistically significant difference (with a marginal statistical power) was found between the *Controller+Trigger* method and *Controller+Grab* method in the position errors, where *Controller+Trigger* was more accurate ($p = 0.015, CI = 0.11 - 2.67, 1 - \beta = 0.42$). There were no statistically significant differences in the orientation errors.

### 5.3. Subjective Data

#### 5.3.1. Evaluations of the Conditions

The interaction methods were subjectively evaluated using five questions; see Table 2. The distribution of subjective rating for those questions are visualized in Figure 5.

For the permutation test, the order of the participant's answers was used as the statistical measure because this rating was performed on an ordinal scale. Computing the differences between the participant's answers for the compared interaction methods, the measure was $+1$ if the difference was positive, $-1$ if the difference was negative and 0 if the answers were the same. The overall measure was then the sum of all participants' numbers.

For the question of participants' confidence (S1), the *HandOnly* method was given statistically significantly lower values ($p < 0.005$) than the *Controller+Trigger* or the *Controller+Grab* method. However, in both cases, the statistical power of the test was very low ($1 - \beta < 0.30$). For the tiredness (S3), the *HandOnly* was valued as being statistically significantly more tiring than *Controller+Grab* ($p = 0.016$), but again with very low statistical power ($1 - \beta = 0.16$).

We analyzed the free-form answers to questions S4 and S5 separately, trying to find common themes. For these two questions, there were no significant differences in the numerical answers.

The *HandOnly* method felt natural because then there is no need to hold a controller but also unnatural, as there were often problems in grabbing or moving the objects. The grabbing/pinching operations sometimes did not work as expected, which frustrated the participants. The *Controller+Trigger* method felt natural because it is a familiar thing and works reliably and accurately, but was felt to be unnatural because of the need to use the trigger. The *Controller+Grab* felt natural because of holding something real in the hand and the grabbing action. On the other hand, some people felt it to be unnatural because the controller was needed.

The *HandOnly* felt suitable for daily use, as it did not need the controller, was playful and easy. On the other hand, it felt too difficult and tiring, and not very accurate. The *Controller+Trigger* felt suitable for daily use, as it was easy to approach. The *Controller+Grab* felt suitable, as it was easy to use and gave more freedom than *Controller+Trigger*. However, the hand grab was not as clear and accurate as using a trigger.

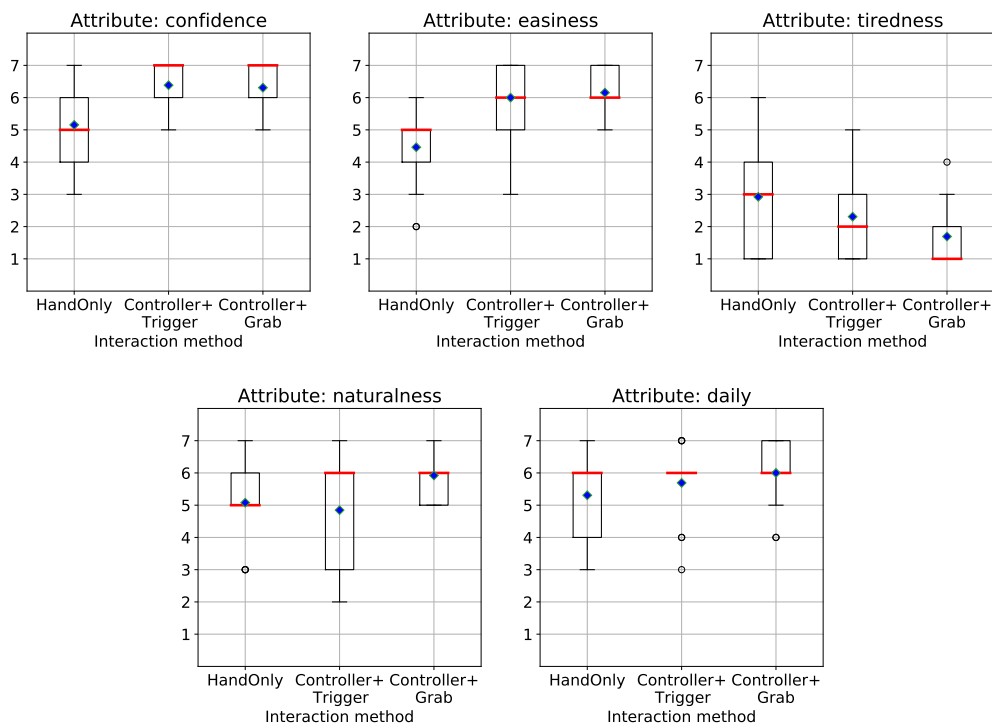

**Figure 5.** The distributions of the subjective ratings for different attributes.

### 5.3.2. Ranking of the Conditions

The interaction methods were ranked in three different dimensions (see Table 3): the most liked, easiest to adopt and having the most development potential. The participants were asked to give a ranking (1st, 2nd, and 3rd) to each method. The frequency of the ranks for each condition are listed in Tables 4–6.

**Table 4.** The number of mentions when asking the participants to rank the interaction methods as the most liked, the second and the least liked.

| Condition | Ranking | | |
|:---:|:---:|:---:|:---:|
| | **1st** | **2nd** | **3rd** |
| *HandOnly* | 2 | 1 | 9 |
| *Controller+Trigger* | 2 | 7 | 3 |
| *Controller+Grab* | 8 | 4 | 0 |

In the comparisons for the most liked method, the *Controller+Grab* was ranked as statistically significantly being more liked than *HandOnly* ($p < 0.010, 1 - \beta = 0.75$). We did not find any significant differences in the comparisons for the easiest to adopt method or for the most potential method.

The *HandOnly* was liked, as it is a natural method. The *Controller+Trigger* was liked, as it is an easy interaction method. However, the *Controller+Grab* was most often the most liked one, and the following reasons were mentioned: The method was easy and accurate,

and a good balance between the naturalness, reliability, and speed. The method was also described as being interesting and intuitive.

**Table 5.** The number of mentions when asking the participants to rank the interaction methods as the easiest to adopt, the second and the least easy.

| Condition | Ranking | | |
|:---:|:---:|:---:|:---:|
| | **1st** | **2nd** | **3rd** |
| *HandOnly* | 4 | 2 | 6 |
| *Controller+Trigger* | 3 | 5 | 4 |
| *Controller+Grab* | 5 | 5 | 2 |

The *HandOnly* method was described as the easiest to adopt, as it is close to the real act of holding an object and does not require any device in hand. The *Controller+Trigger* method was described as the easiest, as people are, anyway, familiar with button-based actions and there was only one trigger to use. The *Controller+Grab* method was described as the easiest, as grasping was natural, and the system was robust. The *Controller+Grab* method also provided haptic feedback as the device was held in the hand.

**Table 6.** The number of mentions when asking the participants to rank the interaction methods as having the most potential, the second and the least potential.

| Condition | Ranking | | |
|:---:|:---:|:---:|:---:|
| | **1st** | **2nd** | **3rd** |
| *HandOnly* | 9 | 1 | 2 |
| *Controller+Trigger* | 1 | 3 | 8 |
| *Controller+Grab* | 2 | 8 | 2 |

The *HandOnly* was seen to have the most potential, as it is the most natural method (with no devices needed), and participants believe that there are possibilities to improve the technology. The *Controller+Trigger* was seen to have the most potential, as the alternatives (the hand-based methods) would be more difficult. The *Controller+Grab* method was seen to have the most potential, as it combines the realistic hand grasp and the device to hold on to.

5.3.3. General Comments

In the end, we offered the participants an opportunity to comment on any specific detail that they felt to be important. One participant mentioned the problems with *HandOnly*, as the technology was not reliable enough. Overall, *Controller+Grab* was seen as the most accurate.

**6. Discussion**

Looking back to the research questions RQ1 and RQ2 (Section 3.1), we found two cases where there were statistically significant differences in the completion times and placement accuracy (research question RQ1). The *HandOnly* method was found to be statistically significantly slower than the two other methods, *Controller+Trigger* and *Controller+Grab*. However, for the test of positioning accuracy, the statistical power was too low, and the statistically significant difference in the test needs to be discarded. No other significant differences were found in the objective measures. The results also confirm Hypothesis H2 (task completion time is shorter for *Controller+Trigger* than for *HandOnly*; Section 3.1).

The results are in line with related earlier studies, e.g., [9–11,13,38] where hand tracking compared unfavorably against the regular controllers for object manipulation tasks. For

example, Caggianese et al. [9] reported that completion times were significantly faster with the controller than with the hand interface, and Gusai et al. [10] concluded that controllers give better performance in accuracy and in usability.

The slow pace of the *HandOnly* method is at least partially explained by the hand pose recognition issues. The current technology cannot always reliably recognize when the user is picking the object or when he/she is releasing it. This sometimes led to repeated trials of object manipulation and, therefore, more time spent on the task. Additionally, the object release was sometimes difficult, as the participant could not say exactly when the system would recognize the release act, and the object might "jump" from an intended position if the hand pose slightly changed at the recognized release time (reported as a "sticky hand" effect by Navarro and Sundstedt [13]). The release recognition problem may also have been the reason for the positioning accuracy difference between the *Controller+Trigger* method and the *Controller+Grab* method (participants commented "...the most difficult to discern when I had successfully released an object", "Releasing accurately was difficult", and "Releasing object felt uncertain").

For the participant preferences (research question RQ2), we found three cases (two related to subjective attributes and one related to ranking) where there were statistically significant differences between the interaction methods. However, the results for subjective attributes had very low statistical power .

The lower values of user confidence for *HandOnly* are most probably caused by the hand pose recognition problems. The participant had difficulties picking up the cubes, which naturally leads to lower confidence. The evaluation of the *HandOnly* method being more tiring may be because it took more time and (sometimes several) repeated efforts to get the task done. When using *HandOnly*, the arm moves and poses do not obviously extend farther or wider than for the other methods (for the extent of hand tracking, see [61]).

The interaction methods were also ranked by the participants in three dimensions: the most liked, easiest to adopt and having the most development potential (see Table 3). For these rankings only the *Controller+Grab* method was ranked as statistically significantly being more liked (R1) than the *HandOnly* method. The results also show that Hypothesis H1 (the *HandOnly* would be preferred over the *Controller+Trigger*) was not confirmed. For the other ranking questions, there were no significant differences between the interaction methods. It is noteworthy that Figueiredo et al. [44] reported that hand tracking was liked more than the controller, even though hands were not as precise as the controllers. Their study showed that some participants liked hands, as they were intuitive to use and did not require any learning, while some participants with more experience with the controller did not feel that there was a high learning curve for the controllers. In addition, some participants in [44] reported feeling a higher level of immersion with hand tracking.

The *Controller+Grab* method was the most liked because of many reasons: the method was easy and accurate, and a good balance between naturalness, reliability, and speed. The *HandOnly* method was liked because it felt natural, and the *Controller+Trigger* because it was easy to use; the *Controller+Grab* had good qualities of both. In the comments, there were no common ideas on what feature would make an interaction method easy to adopt. The *HandOnly* was seen as having the most opportunities for improvement, despite the many problems with its use. *HandOnly* is felt to be a natural way of interaction, which Navarro and Sundstedt [13] described as an interaction concept of "using your own body". The participants commented: "Very much enjoyed manipulating objects without the controller", and "The *HandOnly* felt still the more difficult to control ...but I would see [*HandOnly*] has the most potential, ...".

## 7. Conclusions

Earlier studies reported hand tracking as an unreliable interaction technique in comparison to controllers for direct object manipulation in virtual reality. In this study, we evaluated three interaction techniques for direct manipulation considering the tracking approach and action for object pickup. This evaluation revealed that there is a trade-off

between task accuracy, task completion time and naturalness for user preference of interaction technique. The contributions of this paper are (1) an evaluation of three interaction techniques for direct manipulation of a rigid object, and (2) finding the trade-off between different factors for direct manipulation techniques.

**Author Contributions:** Conceptualization, J.K., H.M. and S.K.K.; methodology, J.K.; software, S.K.K.; validation, J.K., H.M. and S.K.K.; formal analysis, J.K.; investigation, J.K.; resources, R.R.; data curation, J.K.; writing—original draft preparation, J.K.; writing—review and editing, J.K., S.K.K., H.M., J.J. and R.R.; visualization, J.K.; supervision, R.R.; project administration, R.R.; funding acquisition, R.R. and J.J. All authors have read and agreed to the published version of the manuscript.

**Funding:** This work was funded by Business Finland, project Digital and Physical Immersion in Radiology and Surgery (decision number 930/31/2019).

**Institutional Review Board Statement:** The study was conducted according to the guidelines of the Declaration of Helsinki. Ethical review and approval were waived for this study, due to nature of the experimental tasks and the participant population.

**Informed Consent Statement:** Informed consent was obtained from all subjects involved in the study.

**Data Availability Statement:** The data collected during the study are available upon request.

**Acknowledgments:** We would like to thank the people in our laboratory for help in developing the experimental system.

**Conflicts of Interest:** The authors declare no conflict of interest. The funders had no role in the design of the study; in the collection, analyses, or interpretation of data; in the writing of the manuscript; or in the decision to publish the results.

## Abbreviations

The following abbreviations are used in this manuscript:

| | |
|---|---|
| VR | Virtual Reality |
| HMD | Head-Mounted Displays |
| 3D | Three dimensional |
| 2D | Two dimensional |

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
