# Peer review of "Trade-Off between Task Accuracy, Task Completion Time and Naturalness for Direct Object Manipulation in Virtual Reality"

_mti, doi:10.3390/mti6010006_

Round 1
Reviewer 1 Report
Efficient object manipulation is a key challenge in HMI. Authors describe a hybrid interaction technique which combines the use of a controller with hand gestures for object manipulation tasks. They rightfully identify that task accuracy and task completion time can impact the technique used but they fall short on explaining the intrinsics of these two KPIs. This is a good research work but in its current form it contains a few flaws that must be addressed first. First, the literature is very incomplete. There is a vast literature in object manipulation described in most VR courses that is missing in this study, and that focuses on improving task accuracy and task completion. Key techniques include direct (actually touching the object) and indirect (many different ways of dealing with accuracy-speed) object manipulation for hand interaction, and address some of the issues actually reported by the users in this study. In this regard, I suggest the authors to read the work of Mark Billinghurst. Another point that is missing in this research is the use of haptic gloves and force feedback devices, which are used in industry for very high precision control and safety. Authors compare their work to 2D interfaces but at least in the medical and industrial domain a lot has changed in the past years. In particular, many hospitals are now experimenting and some are actually using HoloLens for planning and executing many tasks that require objects and data manipulation. It is not virtual reality, but we began now to see similar works using the hands interaction of Oculus. This leads me to question the sentence: P1 L2: "A handheld controller is the most common interaction 3 method for direct object manipulation in virtual reality environments" Is there any evidence of this claim ? I would suggest rewriting it, unless authors have data that can support this claim. Another aspect that needs to be addressed by the authors is the manipulation of rigid vs deformable bodies, and multi-object interaction. The authors only make references to works that study the manupulation of a single 6DoF point. However, when two hands must be used for manipulation, hybrid techniques introduce an overhead for the modality switch, which should be considered. Other remarks: P2 L50 Is textit part of the text or is it a LaTeX command.Author Response
Efficient object manipulation is a key challenge in HMI. Authors describe a hybrid interaction technique which combines the use of a controller with hand gestures for object manipulation tasks. They rightfully identify that task accuracy and task completion time can impact the technique used but they fall short on explaining the intrinsics of these two KPIs. This is a good research work but in its current form it contains a few flaws that must be addressed first. First, the literature is very incomplete. There is a vast literature in object manipulation described in most VR courses that is missing in this study, and that focuses on improving task accuracy and task completion. Key techniques include direct (actually touching the object) and indirect (many different ways of dealing with accuracy-speed) object manipulation for hand interaction, and address some of the issues actually reported by the users in this study. In this regard, I suggest the authors to read the work of Mark Billinghurst.
We have rewritten and restructured the background section, adding several references to prior work and description to be more comprehensive. We have also edited the Introduction section to better describe the targets of the study. The background section has been divided to several subsections to structure the content better. We added one paragraph of indirect interaction methods and of our emphasis on using direct interaction methods.
Another point that is missing in this research is the use of haptic gloves and force feedback devices, which are used in industry for very high precision control and safety.
We have added some background text and several references about the use of data gloves and force feedback devices. These devices add accuracy and reliability in tracking, but simultaneously they add technical complexity and make the use less natural (as the user needs to hold more accessories or the gloves are heavier and physically larger due to exoskeletons and actuation systems for force feedback).
Authors compare their work to 2D interfaces but at least in the medical and industrial domain a lot has changed in the past years. In particular, many hospitals are now experimenting and some are actually using HoloLens for planning and executing many tasks that require objects and data manipulation. It is not virtual reality, but we began now to see similar works using the hands interaction of Oculus. This leads me to question the sentence: P1 L2: "A handheld controller is the most common interaction 3 method for direct object manipulation in virtual reality environments" Is there any evidence of this claim ? I would suggest rewriting it, unless authors have data that can support this claim.
While the AR and VR devices are available and gradually adopted by medical field, the medical experts are still heavily using 2D interfaces in their everyday work as these are well known, tried, and tested. The VR technologies are developing and they will gradually become as common as the 2D systems are currently, but this is still not so. We added some examples of existing research for using XR technologies in medicine, along with explaining the present state of their use.
About the common interaction methods, the most common use case for VR devices currently is entertainment and commerce where the interaction requirements are different than in medical field.
We have edited the abstract slightly as we don't have evidence to support the claim on line L2 (which was based on observations of the present commercial devices).
Another aspect that needs to be addressed by the authors is the manipulation of rigid vs deformable bodies, and multi-object interaction. The authors only make references to works that study the manipulation of a single 6DoF point. However, when two hands must be used for manipulation, hybrid techniques introduce an overhead for the modality switch, which should be considered.
The target application was a manipulation of rigid single object, that is rotated and moved around (for observation and accurate positioning). We have edited the introduction and background text to clarify these aspects.
Other remarks: P2 L50 Is textit part of the text or is it a LaTeX command.
Thank you for the observation. That was a typing mistake (an incomplete LaTeX commad) and has been corrected.
Reviewer 2 Report
This paper presents a user study comparing direct manipulation methods in VR. Specifically, the study compares along two axes: tracking method (either hand tracking using a Leap motion or controller tracking) and the selection action (either a hand gesture or trigger button). The results show that HandOnly was slower than the other methods with no differences in accuracy. In subjective metrics, the controller+grab technique was more liked than handonly, likely due to its ease of use, accuracy, and balance between naturalness and speed.
Overall, the paper is generally clear and well written; however, there are some specific areas for improvement. The background sections seems particularly brief for a journal article and is missing relevant citations (noted below). Moreover, it simply reads as a list of prior results. I would have expected a bit more discussion of how they relate to each other, justifications given for why these results were found and how that might affect the current study, etc. There are also almost no prior work listed comparing hand tracking with a data glove to a controller, which seems relevant.
Yi-Jheng Huang, Kang-Yi Liu, Suiang-Shyan Lee & I-Cheng Yeh (2021) Evaluation of a Hybrid of Hand Gesture and Controller Inputs in Virtual Reality, International Journal of Human–Computer Interaction, 37:2, 169-180, DOI: 10.1080/10447318.2020.1809248
Lee, K. Park, J. Lee and K. Kim, "User Study of VR Basic Controller and Data Glove as Hand Gesture Inputs in VR Games," 2017 International Symposium on Ubiquitous Virtual Reality (ISUVR), 2017, pp. 1-3, doi: 10.1109/ISUVR.2017.16.
Khundam, C.; Vorachart, V.; Preeyawongsakul, P.; Hosap, W.; Noël, F. A Comparative Study of Interaction Time and Usability of Using Controllers and Hand Tracking in Virtual Reality Training. Informatics 2021, 8, 60. https://doi.org/10.3390/ informatics8030060
Pham, Duc-Minh, and Wolfgang Stuerzlinger. "Is the pen mightier than the controller? a comparison of input devices for selection in virtual and augmented reality." 25th ACM Symposium on Virtual Reality Software and Technology. 2019.
For the user study, only 12 participants is a bit low and may have led to the low statistical power for some metrics. An average of two years prior VR experience among the participants was higher than I expected and may have biased them towards the controller options. Were any statistical tests run to see if prior experience was a factor in performance?
Finally, the last issue concerns novelty and impact, which seem somewhat low. There have been many prior studies comparing hand vs controller input. Although, as the authors note, there is not universal agreement on which is better, I am not sure that this study provides conclusive evidence one way or the other. It is valuable in terms of an additional data point for this task, but ultimately, what this type of study tends to show is that the leap motion tracking is not as robust as we would like. However, the hybrid grabbing technique that makes use of the capacitive touch sensors on the Valve Index is a nice work around at the moment. Given the simplicity of the taxonomy presented in table 1, it seems to be a bit of an overclaim for a "systematic evaluation of design space of interaction techniques for direct manipulation".
There are some minor spelling/grammar errors to correct:
- P1, L11: Add a period after grab
- P1, L12: Add a comma after techniques
- P2, L46: if should be of
- P2, L50: Add a \ before textit for “Controller+trigger”
- P5, L186: Remove the last e in After
Author Response
Overall, the paper is generally clear and well written; however, there are some specific areas for improvement. The background sections seems particularly brief for a journal article and is missing relevant citations (noted below). Moreover, it simply reads as a list of prior results. I would have expected a bit more discussion of how they relate to each other, justifications given for why these results were found and how that might affect the current study, etc. There are also almost no prior work listed comparing hand tracking with a data glove to a controller, which seems relevant.
We have rewritten and refocused the background section as indicated by the reviewer and included new references. We added the suggested references and several other references. We added more discussion of the prior results. We added references and a short discussion of data glove use.
For the user study, only 12 participants is a bit low and may have led to the low statistical power for some metrics. An average of two years prior VR experience among the participants was higher than I expected and may have biased them towards the controller options. Were any statistical tests run to see if prior experience was a factor in performance?
We agree that the number of participants was somewhat low and may have affected the statistical power. The COVID-19 situation affected our ability to conduct experiments that required close personal contacts and we compromised by limiting the number of people. Based on our analysis, the main results are valid with the included number of participants.
The volunteer participants were searched from a population that have had an opportunity to try new interaction technologies, even if not necessarily VR technologies. The most common experience that was reported was 1 year, the group mean was raised to two years mainly by two participants who had used such devices for four and five years, respectively.
We did calculate Pearson's correlation, which didn't indicate any correlation between prior experience and performance.
Finally, the last issue concerns novelty and impact, which seem somewhat low. There have been many prior studies comparing hand vs controller input. Although, as the authors note, there is not universal agreement on which is better, I am not sure that this study provides conclusive evidence one way or the other. It is valuable in terms of an additional data point for this task, but ultimately, what this type of study tends to show is that the leap motion tracking is not as robust as we would like. However, the hybrid grabbing technique that makes use of the capacitive touch sensors on the Valve Index is a nice work around at the moment. Given the simplicity of the taxonomy presented in table 1, it seems to be a bit of an overclaim for a "systematic evaluation of design space of interaction techniques for direct manipulation".
We agree with the reviewer in that the results mostly confirm the problems revealed by earlier work related with the hand tracking use in object manipulation. However, we show that the combination of controller-based tracking and the hand gesture based selection was a nice compromise that was liked by the participants. Also, we emphasize the tradeoff that was revealed between three different relevant factors: accuracy, completion time and naturalness.
Separately, even as we consider the study well done, we removed the word 'systematic' from the text not to overstate the type of evaluation conducted.
There are some minor spelling/grammar errors to correct:
Finally, we corrected all the listed (five) spelling mistakes as pointed out by the reviewer, as well as performed a thorough language check for the final submission.
Reviewer 3 Report
This paper presents a comparative study among three interaction techniques: HandOnly, Controller+Trigger and Controller+ Grab.
The paper is well written and the research methodology is almost correct.
My only concern is related to questions used to gather subjective feedbacks. Why did not Authors use a standard questionnaire such as SUS or NASA-TLX? These questionnaires are well consolidated tools to measure system usability; moreover, NASA-TLX explicitly provides a score for cognitive and physical workload. I'd like Authors were able to perform more tests using a standard questionnaire to gather users' feedback; maybe, custom questions listed in Table 3 could be added.
Author Response
My only concern is related to questions used to gather subjective feedbacks. Why did not Authors use a standard questionnaire such as SUS or NASA-TLX? These questionnaires are well consolidated tools to measure system usability; moreover, NASA-TLX explicitly provides a score for cognitive and physical workload. I'd like Authors were able to perform more tests using a standard questionnaire to gather users' feedback; maybe, custom questions listed in Table 3 could be added.
We selected a short list of (our own) questions to limit the number of questions and to simplify the questionnaires that the participants need to fill during the experiment. In retrospect we probably should have selected one or both standard sets, but unfortunately, we cannot make any changes to this experiment.
Round 2
Reviewer 1 Report
The authors have addressed the suggestions of the previous review
Author Response
Thank you for the comments and suggestions in the earlier review. We have read through the manuscript for some edits in the language.
Reviewer 2 Report
This revision of the paper is significantly improved. In particular the reorganization and extension of the background section provides more context. The authors have also weakened some of the over-claims as noted in the original review. Overall, the impact is still not exemplary, but the work does present an important confirmation of different factors to consider when designing direct manipulation interfaces. The authors have satisfactorily addressed my concerns, and I would recommend publication after a minor spelling/grammar check.
Author Response
Thank your for the comments and suggestions in the earlier reviews. We have read through the manuscript for several corrections in the language.